# Chemical Profiling of *Drimys granadensis* (Winteraceae) Essential Oil, and Their Antimicrobial, Antioxidant, and Anticholinesterase Properties

**DOI:** 10.3390/plants13131806

**Published:** 2024-06-30

**Authors:** Luis Cartuche, Camila Vallejo, Edison Castillo, Nixon Cumbicus, Vladimir Morocho

**Affiliations:** 1Departamento de Química, Técnica Particular de Loja (UTPL), Calle París s/n y Praga, Loja 110107, Ecuador; svmorocho@utpl.edu.ec; 2Carrera de Bioquímica y Farmacia, Universidad Técnica Particular de Loja (UTPL), Calle París s/n y Praga, Loja 110107, Ecuador; 3Departamento de Ciencias Biológicas y Agropecuarias, Universidad Técnica Particular de Loja (UTPL), Calle París s/n y Praga, Loja 110107, Ecuador; nlcumbicus@utpl.edu.ec

**Keywords:** *Drimys granadensis*, essential oil, anticholinesterase, GC/MS, DPPH/ABTS, MIC, *Listeria monocytogenes*

## Abstract

A complete and comprehensive chemical and biological study of *Drimys granadensis*, a native Ecuadorian aromatic plant, was conducted. By conventional steam distillation from dried leaves, a yellowish, translucent essential oil (EO) with a density of 0.95 and a refractive index of 1.5090 was obtained. The EO was analyzed by gas chromatography coupled to a mass spectrometer (GC/MS) and an FID detector (GC/FID), respectively. Enantiomeric distribution was also carried out by GC/MS using a chiral selective column (diethyl tert-butylsilyl-BETA-cyclodextrin). The microdilution broth method was employed to assess the antibacterial and antifungal activity of the EO against a panel of opportunistic microorganisms. Antioxidant capacity was measured using diphenyl picryl hydrazyl (DPPH) and azino-bis 3-ethylbenzothiazoline-6-sulfonic acid (ABTS) radicals. Finally, the inhibitory potential of the EO against acetylcholinesterase was also valued. Sixty-four chemical compounds, constituting 93.27% of the total composition, were identified, with major components including γ-muurolene (10.63%), spathulenol (10.13%), sabinene (5.52%), and δ-cadinene (4.22%). The characteristic taxonomic marker of the *Drimys* genus, Drimenol, was detected at very low percentages (<2%). Two pairs of enantiomers ((1S,5R)-(+)-α-pinene/(1S,5S)-(–)-α-pinene; (1S,5R)-(+)-β-pinene/(1S,5S)-(–)-β-pinene) and one pure enantiomer (1R,4S)-(–)-camphene were identified. Regarding antimicrobial potency, the EO exhibited a significant moderate effect on *Listeria monocytogenes* with a minimal inhibitory concentration (MIC) value of 250 µg/mL, while with the remaining microorganisms, it exerted less potency, ranging from 500 to 2000 µg/mL. The EO displayed moderate effects against the ABTS radical with a half scavenging capacity of 210.48 µg/mL and no effect against the DPPH radical. The most notable effect was noticed for acetylcholinesterase, with a half inhibition concentration (IC_50_) of 63.88 ± 1.03 µg/mL. These antiradical and anticholinesterase effects hint at potential pharmacological applications in Alzheimer’s disease treatment, although the presence of safrole, albeit in low content (ca. 2%), could limit this opportunity. Further in vivo studies are necessary to fully understand their potential applications.

## 1. Introduction

The Winteraceae family comprises approximately 130 species across seven genera, with the greatest species diversity found in the Pacific (*Pseudowintera*, *Zygogynum*), Australia (*Bubbia*, *Tasmannia*), New Guinea (*Belliolum*, *Bubbia*, *Zygogynum*, *Tasmannia*), and Madagascar (*Takhtajania*). Only the genus *Drimys* is found in South America [1,2,3]. The term “drimys” originates from the Greek word denoting pungent or spicy, alluding to the distinct flavor of its bark and leaves [4].

The genus *Drimys* primarily inhabits South America, particularly the Andean regions of Colombia, Ecuador, Peru, and Venezuela, where it thrives in cloud forests and humid montane forests [5]. These plants are characterized by their large shrubby form, featuring reddish stems and young glaucous branches. Their leaves boast dark green hues and have wrinkled, flattened petioles. Flowers are clustered in umbels near the branches, each supported by peduncles carrying one to six flattened flowers. Typically, the sepals number two to three, while the petals exhibit shades ranging from yellowish to reddish [1]. In *Drimys*, the presence or absence of conspicuous glands on the stamen connectives has been used as a taxonomic characteristic and is considered a synapomorphy for the clade that includes *Drimys angustifolia*, *D. brasiliensis*, *D. granadensis*, and *D. roraimensis* (Northeastern clade). However, detailed anatomical studies of stamens and carpels have only been conducted for *D. winteri* [2].

The *Drimys* species harbor a range of natural active compounds renowned for their therapeutic properties. For instance, the infusion derived from cinnamon tree bark has historically been valued for its high vitamin C content, effectively combating scurvy [6]. Moreover, the essential oils extracted from the leaves and green fruits of *Drimys granadensis* have been employed in treating a spectrum of ailments, including renal, rheumatic, and lumbar disorders. Additionally, they exhibit analgesic and anti-inflammatory properties, while also serving as stimulants for the body. Notably, in traditional medicine, the leaves, bark, and fruits of *Drimys* are the most commonly utilized parts [7].

Regarding its chemical composition, *D. granadensis* species are notably abundant in sesquiterpenes, featuring compounds such as germacrene D, sclarene, longiborneol acetate, drimenol, Z-(*β*)-ocimene, α-pinene, and (*β*)-elemene [8]. Additionally, monoterpene compounds like 4-terpineol, sabinene, α-pinene, and limonene are also present [9]. In related species, like *Drimys winteri*, the occurrence of sesquiterpnes like drimane such as Polygodial, 1-*β*-(*p*-methoxycinnamoyl) Polygodial, Mukaadial, and Drimanial has been reported, exhibiting remarkable antinociceptive, anti-inflammatory, and antiallergic actions [10]. Likewise, *from* the *Drimys brasiliensis* essential oil, a predominance of sesquiterpenes such as Hinesol, *β-Eudesmol*, *α-Eudesmol*, *Elemol*, *Epi-Cyclocolorenone*, *α-pinene* and *safrole* [11], as well as Germacrene D, Bicyclegermacrene, epi-α-Cadinol, α-Cadinol, Drimenol, (*E*)-nerolidol, and spathulenol, among others, was reported [12,13].

Despite the extensive study of the chemical composition and potential therapeutic benefits of essential oils from the genus *Drimys*, there remains a dearth of information on the chemical composition of the essential oil extracted from dehydrated leaves of *D. granadensis* in Ecuador. Additionally, updated literature on the *Drimys* species is scarce. Thus, this research aims to increase and update the data on the chemical composition and enantiomeric distribution of the essential oil derived from *Drimys granadensis*, as well as to evaluate various biological properties not previously tested, including anticholinesterase and antioxidant activities, and to support the microdilution broth method, the antimicrobial effect observed by Gaviria et al. [8], who reported preliminary antibacterial activity via the disc diffusion method.

## 2. Results and Analysis

### 2.1. General Properties of Essential Oil

Essential oils were obtained by steam distillation of the dried leaves of *Drimys granadensis*, and the sample was obtained as a yellowish, viscous, translucent liquid. A volume of 0.12 mL was obtained, representing a very low yield of 0.12 ± 0.064%. Regarding the physical properties, the EO displayed a density of 0.95 ± 0.02 and a refractive index of 1.5090 ± 0.06.

### 2.2. Chemical Composition of EO from Leaves of D. granadensis

The essential oil (EO) extracted from the dried leaves of Drimys granadensis through steam distillation was subjected to analysis via gas chromatography coupled to mass spectrometry (GC-MS) for qualitative assessment and a flame ionizing detector (GC-FID) for quantitative evaluation. A DB-5 ms apolar column was employed to identify volatile compounds. The results revealed 64 compounds, constituting 93.27% of the total oil composition. Among these, γ-muurolene emerged as the primary chemical compound at 10.63%, followed by spathulenol (10.13%), sabinene (5.52%), and δ-cadinene (4.22%). Other compounds (<4%) included α-cadinol (3.58%), bicyclogermacrene (2.50%), β-pinene (2.46%), safrole (2.04%), γ-terpinene (1.97%), and α-pinene (1.96%), to cite some. The complete chemical characterization is displayed in Table 1. Sesquiterpenes predominated in this species, comprising 65.43% of the identified compounds, categorized into hydrocarbon sesquiterpenes (32.267%) and oxygenated sesquiterpenes (33.165–32.542%). The gas chromatography (GC-MS) profile is illustrated in Figure 1.

### 2.3. Enantioselective GC Analysis of Essential Oil

Enantioselective analysis was conducted using the MEGA-DEX DET-Beta column (diethyl tert-butylsilyl-BETA-cyclodextrin) as a chiral selector. Notably, *Drimys granadensis* exhibited chiral compounds such as (–)-α-pinene, (–)-β-pinene, and (–)-camphene, showcasing significant enantiomeric excess. Table 2 provides a comprehensive overview of the calculated retention indices, literature retention indices, enantiomeric distribution, and enantiomeric excess for these compounds.

### 2.4. Antimicrobial and Antifungal Activity of D. granadensis

The antibacterial and antifungal activity of the essential oil derived from *D. granadensis* leaves was assessed using the microdilution broth method. Positive controls for antimicrobial activity included ampicillin, ciprofloxacin, erythromycin, and amphotericin B, as outlined in Table 3.

*D. granadensis* demonstrated notable activity against *Listeria monocytogenes* and *S. aureus*, exhibiting MIC values of 250 µg/mL and 500 µg/mL, respectively. However, there was no significant effect observed against the remaining microorganisms, with MIC values ≥ 1000 µg/mL (Table 3).

### 2.5. Antioxidant Capacity

The antioxidant capacity of *D. granadensis* EO was tested against two stable radicals, DPPH and ABTS. Additionally, the antiradical capacity was expressed as TEAC by comparing it with a standard curve of Trolox (a synthetic analog of vitamin E). The data obtained are shown in Table 4.

### 2.6. Anticholinesterase Potential

Figure 2 illustrates the inhibitory effect of the essential oil, expressed as residual enzymatic activity. The calculated IC_50_ value obtained from the corresponding curve for D. granadensis EO was 63.88 ± 1.03 µg/mL. As a reference, the anticholinesterase potential of the positive control, donepezil, displayed an IC_50_ value of 12.40 ± 1.35 nM, serving solely as an internal control for the assay. These results indicate that *D. granadensis* EO possesses a moderate anticholinesterase effect relative to donepezil, a well-known drug with cholinesterase inhibitory activity, yet demonstrates strong potential compared to other essential oils.

## 3. Discussion

In a previous study on the composition of *Drimys granadensis* essential oil (EO) in Colombia, 85 compounds were identified, representing 98.6% of the EO. Major compounds extracted from the leaves included germacrene D (14.7%), sclarene (9.5%), α-cadinol (7.3%), longiborneol acetate (6.3%), drimenol (4.2%), and (Z)-β-ocimene (4.2%). Additional compounds (<5%) such as α-pinene (3.2%) and β-elemene (2.7%) were also present. Similarly, sesquiterpenes were the main chemical group, comprising 73.7% of the chemical composition, divided between hydrocarbon sesquiterpenes (36.1%) and oxygenated sesquiterpenes (37.6%) [8]. Compared to our study, twenty-eight compounds were identified and matched, but in different proportions, including α-pinene, camphene, sabinene, β-pinene, myrcene, α-terpinene, γ-terpinene, terpinolene, terpinen-4-ol, α-terpineol, safrole, α-cubebene, eugenol, α-ylangene, β-bourbonene, β-elemene, β-gurjunene, cis-muurola-3,5-diene, α-humulene, γ-muurolene, germacrene D, δ-cadinene, α-cadinene, cubenol, α-cadinol, eudesma-4(15),7-dien-1b-ol, and drimenol.

A similar study conducted in Costa Rica by Cicció [9] identified 42 compounds from the essential oil (EO) of both the leaves and fruits of *D. granadensis*. 4-terpineol (21.9%) was the major compound extracted from fresh leaves, followed by sabinene (16.6%) and germacrene D (10.2%). Additional compounds (<2%) such as δ-cadinene (1.8%) and safrole (1.2%) were also present. Monoterpenes constituted the predominant chemical group in this species, representing 71.5% of the identified compounds, divided into hydrocarbon monoterpenes (45.8%) and oxygenated monoterpenes (25.7%). Notably, 15 compounds were shared among these samples, albeit in varying amounts, including δ-cadinene, sabinene, α-thujene, germacrene D, α-pinene, β-pinene, γ-terpinene, safrole, terpinen-4-ol, α-terpineol, drimenol, α-cubebene, β-bourbonene, β-elemene, and α-humulene. In contrast, the major components in immature fruit samples were germacrene D (23.4%), drimenol (10.0%), and δ-cadinene (5.0%), accounting for 54.4% of the identified sesquiterpenes, divided into hydrocarbonated (41.8%) and oxygenated categories.

In a study on a similar species, *D. brasiliensis*, collected in Brazil, the chemical characterization of its essential oil (EO) identified 62 compounds, with 28 of them bearing similarities to our research but in significantly different proportions. Notably, major compounds included α-pinene (18.4%), limonene (13.7%), sabinene (9.6%), β-pinene (8.6%), terpinen-4-ol (6.2%), and bicyclogermacrene (5.6%). Monoterpenes also dominated as the majority group, comprising 63.93% of the total oil composition [12].

The proportions and compositions of the essential oils from different *Drimys granadensis* populations vary considerably from the results obtained in the present investigation. Notable differences were found for *Drymis granadensis* from Colombia, Costa Rica, and Ecuador, including variation in the amount of chemicals detected, such as 85, 42, and 64 compounds, respectively. The major component reported in Colombia was germacrene D (14%), while Costa Rica had 4-terpineol (21.9%) as the major component. Ecuador’s oil had γ-muurolene and spathulenol at 10% each. Another notable difference includes the dominance of monoterpenes (71.5%) in the EO reported in Costa Rica, while Colombia and Ecuador had similar proportions of sesquiterpenes (73.7% and 65.44%, respectively). These variations in chemical composition are likely influenced by the genetic origin of the plant, phenological stage, climatic conditions, and geographical location of the different populations of Drimys spp., as suggested by Muñoz et al. [16]. Some other factors less considered, such as wind speed, vegetation, soil type, radiation, time of harvesting, as well as chemical conditions like isomerization, cyclization, or dehydrogenation, can be determining factors that alter the occurrence of several metabolites [17,18].

In terms of antimicrobial potential, the essential oil (EO) demonstrated no effect at the maximum tested against Gram-negative bacilli bacteria, including *E. coli* (O157:H7), *P. aeruginosa*, and *S. enterica*, and against *C. albicans*, a pathogenic human yeast. However, a moderate inhibitory effect was observed for *Campylobacter jejuni*, with a MIC of 2000 µg/mL. Conversely, significant effects against Gram-positive cocci were noted for *D. granadensis* EO, with a MIC value of 500 µg/mL for *S. aureus* and a moderate effect with a MIC of 1000 µg/mL against *Enterococcus faecium* and *Enterococcus faecalis*. Notably, a more pronounced effect was observed for *Listeria monocytogenes* (a Gram-positive bacillus) with a MIC value of 250 µg/mL. Following the classification proposed by Van Vuuren and Holl [19], MIC values equal to or higher than 1000 µg/mL are deemed worthy of publication, while values between 101 and 500 µg/mL indicate a strong antimicrobial effect. Likewise, Tamokou et al. [20] proposed a more accurate classification on the cutoffs of MIC values for extracts of edible plants. Considering this, *D. granadensis* EO is significantly active against *L. monocytogenes* because its MIC value lies between 100 and 512 µg/mL. Regarding antifungal activity, results obtained for yeasts such as *Candida albicans* and sporulating fungi like *A. niger* were not as favorable as those observed for Gram-positive bacteria.

In an analysis conducted by Cuervo et al. [21], the bactericidal effect of plant extracts of *D. granadensis* at different solvent polarities demonstrated that, in general terms, the genus *Drimys* exhibits antimicrobial and cytotoxic properties. This antimicrobial effect was also evidenced in species such as *Drimys angustifolia* and *Drimys brasiliensis*, which exerted antiviral activity at concentrations of 156.3 µg/mL and 625 µg/mL, respectively.

It can be concluded that polar to moderately polar components of *D. granadensis* are mainly responsible for conferring this biological potential. It is also postulated that drimane-type sesquiterpenoids possess a wide range of biological activities, including antimicrobial, antibacterial, antifungal, antifeedant, cytotoxic, growth-regulating, and phytotoxic properties. Muñoz et al. [22] assessed the antifungal activity of drimenol and demonstrated that germination of *B. cinerea* is reduced by almost 50% at 40 and 80 ppm. Unfortunately, this capacity could not be detected in our antifungal assay, probably due to the low content of drimenol in our EO (1.91%).

The EO of *Drimys granadensis* demonstrated insignificant scavenging capacity against the DPPH free radical, with a 50% scavenging concentration (SC_50_) of 4181.74 ± 1.47 µg/mL, while exhibiting moderate activity against the ABTS radical, with an SC_50_ of 210.48 ± 1.03 µg/mL. A parallel study on *D. angustifolia* and *D. brasiliensis* essential oils revealed their inability to neutralize the stable DPPH radical, rendering them inactive against this model [23]. Conversely, research on *D. winteri* indicated that the crude extract from fresh fruits displayed significant antioxidant capacity, boasting IC_50_ values of 6.65 µg/mL and 9.5 µg/mL against DPPH and ABTS radicals, respectively [24]. This heightened activity is likely attributed to the abundance of phenolic compounds naturally occurring in the extracts, a characteristic not commonly found in essential oils.

The essential oil of *D. granadensis* displayed moderate anticholinesterase activity, with an IC_50_ value of 63.88 ± 1.03 µg/mL, indicating its pharmacological potential as an inhibitor of the acetylcholinesterase enzyme (AChE). However, due to its rich chemical composition, establishing a direct relationship between the dominant major compounds and the observed anti-AChE effect is challenging.

According to da Silva et al. [25], the bicyclic sesquiterpene *δ*-cadinene (occurring as a major component of the EO of *M. floribunda*) may be responsible for the strong inhibitory effect observed against AChE, with an IC_50_ value of 0.08 µg/mL, as reported in their study. This assertion is further supported by docking studies, demonstrating that the interaction occurs via hydrophobic and van der Waals links with amino acids TRP86, TYR124, PHE295, TYR337, TRP286, and VAL294 present in the native structure of the AChE enzyme, thereby interrupting its catalytic mechanism. δ-cadinene was detected at a concentration of 26.8% in *M. floribunda*, whereas it constituted only 4.22% in our EO of *D. granadensis*. This discrepancy indicates that the activity of the *D. granadensis* EO is not solely attributed to this compound.

In accordance with the research of Kawamoto et al. [26], *β*-elemene, *α*-humulene, and spathulenol were evaluated for their inhibitory activity against AChE. *β*-elemene and *α*-humulene exhibited moderate inhibitory potential with IC_50_ values of 77.2 and 298.2 µM, respectively, while spathulenol showed no inhibitory effect. Despite the low concentrations of *β*-elemene and α-humulene in our essential oil (approximately 1%), they may contribute to the overall inhibitory effect through synergistic mechanisms. However, despite spathulenol being present at a concentration of 10% in the EO of *D. granadensis*, it is evident that it did not participate in the observed inhibitory effect according to this study.

Sabinene, a natural bicyclic monoterpene present at 5% in our essential oil, may exert an inhibitory effect on AChE, as suggested by Menichini et al. [27], who reported a promising inhibitory effect of sabinene (occurring at 4.4% in the EO of *Pimpinella anisoides*) with an IC_50_ value of 176.5 µg/mL. Similarly, according to Miyasawa et al. [28], bicyclic monoterpenes such as (–)-*α*-pinene and (+)-3-carene have demonstrated significant inhibitory effects against AChE, with IC_50_ values of 440 µM and 200 µM, respectively. Although (1S,5S)-(–)-*α*-pinene was detected in our EO with an enantiomeric excess of 77%, its relative abundance is less than 2% of the total chemical composition.

Despite γ-muurolene being another main compound occurring at 10% in our EO, there are no reports about its efficacy against AChE but, it is clear that the main inhibitory effect of the EO of *D. granadensis* against AChE can be attributed to the occurrence and interaction of several bioactive compounds, as cited above.

Despite the remarkable activities displayed by the EO of *D. granadensis*, we should not be careless about the fact that safrole is a well-known carcinogenic compound (in mice), as demonstrated in several studies [29], and is a minority component in the EO (2.04%). Safrole is characterized as a B2 chemical by EPA, listed as a probable human carcinogen, although there is no data supporting its effects on humans. Further studies should be conducted to determine the toxicity and final applications of this EO.

## 4. Materials and Methods

### 4.1. Plant Material

The leaves of *D. granadensis* were collected on 6 October 2023, from many trees, distributed within a distance of about 100 m around a central point, having coordinates of 04°23′08′′ S and 79°07′13′′ W, at an altitude of 2950 m a.s.l. The collection was made with the permission of the Ministry of Environment, Water, and Ecological Transition of Ecuador, with the MAATE registry number, MAE-DNB-CM-2016-0048. Taxonomical identification was carried out by botanist, Dr. Nixon Cumbicus, and the botanical specimens were conserved at the herbarium of the Universidad Técnica Particular de Loja, with the voucher code, 14812.4.2.

### 4.2. Postharvest Treatments

Upon arrival at the laboratory, the leaves were processed and sorted, with those showing signs of deterioration discarded, and then dehydrated in a drying chamber at 34 °C for 24 h.

### 4.3. Essential Oil Extraction

A total of 100 g of dried leaves of *D. granadensis* were subjected to steam distillation using Clevenger equipment for a period of 3 h, and the essential oil obtained by density was collected, dried under anhydrous sodium sulphate, labeled, and stored at −4 °C to avoid the loss of its contents and the alteration of its compounds.

### 4.4. Refraction Index and Density

Two physical properties of the essential oil (EO) were determined: relative density (d20) and refractive index. The relative density was determined according to the international standard AFNOR NF T75-111 (ISO 279:1998 [30]) using an analytical balance (Mettler AC100 model) and a 1 mL pycnometer. The refractive index was measured using an ABBE refractometer (Boeco, Hamburg, Germany) according to the international standard AFNOR NF 75-112 (ISO 280:1998 [31]). Measurements were performed twice at 20 °C.

### 4.5. Essential Oil Composition

#### 4.5.1. Qualitative Analysis

A Thermo Fisher Scientific (Carlsbad, CA, USA) gas chromatograph model TRACE 1310, series 720002174, and a Thermo Fisher Scientific mass spectrometer model ISQ 7000 were used. Data were analyzed via the “Chromeleon 7.3 software”. Full scan mode spectra were recorded in a mass range of 40 to 400 *m*/*z* with a scanning speed of 0.2 scan/s. A 1:1000 dilution of the essential oil (EO) in methylene chloride was used, and 1 µL of this solution was injected in a DB5-MS apolar column (5% phenylmethylpolysiloxane, 30 m × 0.25 mm id, 0.25 μm film thickness, J & W Scientific, Folsom, CA, USA). The GC/MS experiment was run under the following conditions: split mode (40:1), with the oven temperature program starting at 50 °C for 5 min, followed by a ramp of 3 °C/min until it reached 250 °C and held for 3 min. The ion source temperature was set to 230 °C. Ultra-pure helium (from Indura, Guayaquil, Ecuador) served as the carrier gas at a constant flow of 1.00 mL/min. The total run time was 66 min. Compound identification was achieved by comparing mass spectra and linear retention indexes (LRIs) with literature references. LRI of the compounds were determined based on the homologous standard aliphatic hydrocarbons TPH-6RPM of Chem Service (mixture of *n-alkane* C9-C25). The LRI was experimentally determined following the method of Van Den Dool and Krats [32].

#### 4.5.2. Quantitative Analysis

The same gas chromatograph was used as above, equipped with a flame ionization detector (FID). GC-FID analyses were performed using the same method and the same instrumental setup as the GC-MS, described above. It differed only in the gas mixture for the injector, with an airflow of 350 mL/min, nitrogen flow of 40 mL/min, and hydrogen flow of 35 mL/min. The run time was 66 min. The percentage composition of the EO was computed by a normalization method from the GC peak area data and was reported as the mean of three injections.

#### 4.5.3. Enantioselective Analysis

Enantioselective analysis was carried out using GC-MS via the MEGA-DEX DET-Beta column, whose stationary phase was diethyl tert-butylsilyl-BETA-cyclodextrin (dimensions: 20 m × 0.25 mm × 0.25 µm). The operating conditions of the equipment were as follows: the volume to be injected was 1 µL; the carrier gas was helium; and the flow rate was 40 mL/min. The injection method was split, with a maximum temperature of 200 °C and a split ratio of 40:1. The temperature program was as follows: 50 °C for 1 min, followed by a thermal gradient of 200 °C at 2 °C/min. The run time was 96 min.

### 4.6. Antimicrobial Activity

The broth microdilution technique was carried out following the guidelines described by Cartuche et al. [33] and according to the M07, M38, and M45 CLSI guidelines [34,35,36,37]. The antimicrobial inhibitory effect of various agents was evaluated using this method. For this purpose, a group of strains was selected from the American Type Culture Collection (ATCC), recognized for being opportunistic microorganisms responsible for infectious diseases in humans and widely used in antimicrobial studies. Three cocci bacteria such as *E. faecalis* ATCC^®^ 19433, *E. faecium* ATCC^®^ 27270, *S. aureus* ATCC^®^ 25923, four rod-shaped bacteria (*L. monocytogenes* ATCC^®^ 19115, *E. coli* (O157:H7) ATCC^®^ 43888, *P. aeruginosa* ATCC^®^ 10145, *S. enterica* ATCC^®^ 14028), a microaerophilic rod-shaped bacterium (*Campilobacter jejuni* ATCC^®^ 33560) and two fungal strains (*C. albicans* ATCC^®^ 10231, *A. niger* ATCC^®^ 6275) were used as microbial models for the assay. The minimum inhibitory concentration was determined, and the antimicrobial potential of the agents studied was evaluated. Concentrations ranging from 4000 to 31.25 µg/mL were achieved using the double serial dilution method, with a final inoculum concentration of 5 × 10^5^ cfu/mL for bacteria, 2.5 × 10^5^ cfu/mL for yeasts, and 5 × 10^4^ spores/mL for sporulated fungi. Mueller Hinton II (MH II) and Sabouraud broths were utilized as test media for bacteria and fungi, respectively. The *Campylobacter jejuni* culture was activated by adding horse serum at 5% in Tioglycolate medium and incubated under microaerophilic conditions (Campygen sachet, Thermo Scientific) at 37 °C for 48 h. The broth microdilution test was performed in MH II supplemented with 5% horse serum (Thermo). Antimicrobial agents such as ampicillin, criprofloxacin, and erythromycin were used at a concentration of 1 mg/mL and, amphotericin B at 250 µg/mL and included in the assays to validate the sensitivity of the microorganism’s panel. MIC values for positive controls are depicted in Table 3 and compared with the information supplied by supplement M100 from CLSI [37].

### 4.7. Antioxidant Capacity

#### 4.7.1. 2,2-Diphenyl-1-picril Hydrazyl (DPPH) Radical Scavenging Assay

The DPPH antiradical assay method was followed according to the procedure described by Cartuche et al. [33], using the free stable radical 2,2-diphenyl-1-picrylhydrazyl (DPPH). A working solution of DPPH was prepared by dissolving 24 mg of DPPH in 100 mL of methanol and then stabilized with more solvent to an absorbance of 1.1 ± 0.01 at 515 nm (this ensures consistent radical concentration for the assay) in an EPOCH 2 microplate reader (BIOTEK, Winooski, VT, USA). The antiradical interaction was evaluated by absorbance measurement. Different concentrations of EO dissolved in methanol were prepared by the two-fold serial dilution method to reach concentrations ranging from 8000 to 62.5 µg/mL. In brief, 270 µL of the adjusted DPPH working solution and 30 µL of the EO at different concentrations were added to a 96 micro-well plate. The reaction was monitored at 515 nm for 60 min at room temperature. As positive and blank controls, Trolox, and methanol were used, respectively. The results were expressed as SC_50_ (sweep concentration of the radical at 50%). All measurements were performed in triplicate to ensure accuracy.

#### 4.7.2. 2,2-Azino-Bis (3-Ethylbenzothiazoline-6-sulfonic Acid) Radical Scavenging Assay

The analysis for antioxidant power using the cation radical ABTS followed the protocol described by Cartuche et al. [33]. Briefly, a stock radical solution was prepared by combining equal volumes of ABTS (7.4 µM) and potassium persulfate (2.6 µM) in water and stirring for 12 h. Subsequently, a standard solution was prepared by diluting the stock solution in methanol to an absorbance of 1.1 ± 0.02, measured at 734 nm in an EPOCH 2 microplate reader (BIOTEK, Winooski, VT, USA). The antiradical reaction was carried out for 1 h in the dark at room temperature by adding 270 µL of the adjusted ABTS working solution and 30 µL of the essential oil (EO) at the same concentrations prepared earlier to each well. Trolox and methanol were used as positive and blank controls, respectively. Results were expressed as SC_50_ (50% of the radical scavenging concentration).

### 4.8. Anticholinesterase Assay

The acetylcholinesterase inhibitory effect of *D. granadensis* EO was evaluated in vitro following the method described by Ellman et al. [38], with some modifications proposed by Andrade et al. [39]. Briefly, the reaction mixture contained 40 μL of Tris buffer pH 8.0, 20 μL of the EO sample, 20 μL of acetylthiocholine as substrate (ATCh, 15 mM in PBS, pH 7.4), and 100 μL of DTNB as reaction revealer (3 mM in Tris buffer). After a 3-min preincubation period at 25 °C with continuous shaking, 20 μL of acetylcholinesterase (0.5 U/mL) was added to initiate the reaction. Product release was monitored at 405 nm on an EPOCH 2 microplate reader (BIOTEK) at 25 °C for 60 min.

The essential oil (EO) was prepared by dissolving 10 mg in 1 mL of methanol (MeOH). Four additional dilutions (10× dilution factor) were performed to obtain final concentrations of 1000, 500, 100, 50, 50, and 10 μg/mL. The rate of product release was determined by measuring absorbance at 405 nm and generating a standard curve of DTNB and L-GSH at various molar concentrations. Methanol (MeOH), chosen as a non-selective protic solvent, was used as a negative control at a maximum concentration of 10%, without affecting the enzymatic reaction. Donepezil hydrochloride was used as a positive control, showing a calculated IC_50_ value of 12.40 ± 1.35 nM.

## 5. Conclusions

The essential oil extracted from *Drimys granadensis* in Ecuador exhibits a diverse array of chemical compounds, particularly terpenoids, with significant variations in proportion compared to previous studies. It presents, for the first time, a comprehensive examination of the chemical composition, enantiomeric proportions, and antimicrobial, antioxidant, and anticholinesterase properties of this essential oil. Notably, it demonstrates a moderate and selective antimicrobial effect against the Gram-positive bacillus, *Listeria monocytogenes*, alongside a moderate antioxidant and anticholinesterase profile. These findings raise intriguing possibilities for potential pharmaceutical applications of this essential oil, particularly in the realm of adjunct therapies for Alzheimer’s disease. However, further research is imperative to identify the specific compounds responsible for these activities and to assess the toxicity of this essential oil, given its trace amounts of safrole.

## Figures and Tables

**Figure 1 plants-13-01806-f001:**
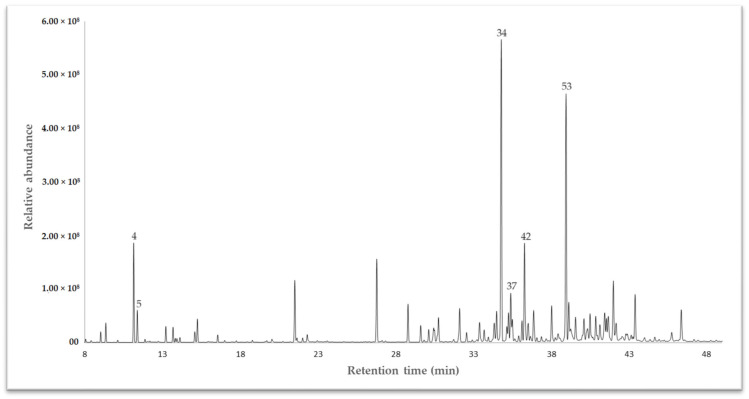
A chromatogram of the essential oil of *Drimys granadensis* from Ecuador, exhibiting some of the main compounds, as depicted in Table 1.

**Figure 2 plants-13-01806-f002:**
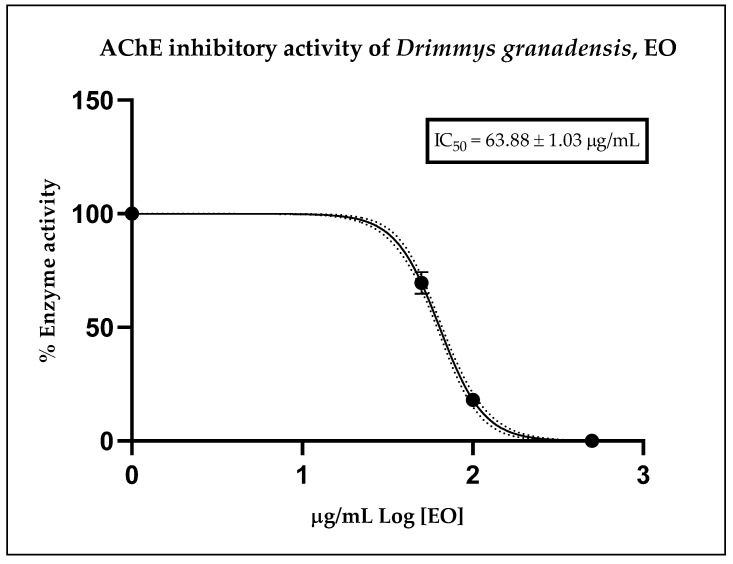
Residual cholinesterase activity exerted by different doses of *D. granadensis* EO. Data were represented as the median of three replicates of three different experiments and were analyzed using a nonlinear regression model. The dot line represents the 95% confidence interval for the fitted curve.

**Table 1 plants-13-01806-t001:** Chemical compounds in the essential oil from the leaves of *D. granadensis*.

No	Compuesto	CRI	LRI	% FID	MF
1	α-Thujene	925	924	0.72	C_10_H_16_
2	α-Pinene	933	932	1.96	C_10_H_16_
3	Camphene	950	946	0.24	C_10_H_16_
4	Sabinene	974	969	5.53	C_10_H_16_
5	β-Pinene	979	974	2.46	C_10_H_16_
6	Myrcene	990	988	0.29	C_10_H_16_
7	α-Phellandrene	1009	1002	0.167	C_10_ H_16_
8	α-Terpinene	1019	1014	1.34	C_10_H_16_
9	ο-Cymene	1029	1022	0.72	C_10_H_14_
10	Limonene	1031	1024	0.32	C_10_H_16_
11	β-Phellandrene	1034	1025	0.72	C_10_H_16_
12	(*Z*)-β-Ocimene	1038	1032	0.02	C_10_H_16_
13	NI	-	-	-	-
14	γ-Terpinene	1061	1054	1.97	C_10_H_16_
15	Terpinolene	1088	1086	0.58	C_10_H_16_
16	2,5-Dimethyl styrene	1097	1099	0.15	C_10_H_12_
17	Terpinen-4-ol	1190	1174	1.10	C_10_H_18_O
18	α-Terpineol	1206	1186	0.15	C_10_H_18_O
19	Safrole	1302	1285	2.04	C_10_H_10_O_2_
20	α-Cubebene	1347	1348	1.27	C_15_H_24_
21	Eugenol	1366	1356	0.49	C_10_H_12_O_2_
22	α-Ylangene	1377	1373	0.55	C_15_H_24_
23	β-Bourbonene	1385	1387	0.59	C_15_H_24_
24	n-Decanoic acid	1388	1382	0.40	C_10_H_20_O_2_
25	β-Elemene	1391	1389	1.18	C_15_H_24_
26	(*E*)-Caryophyllene	1423	1417	1.69	C_15_H_24_
27	β-Gurjunene	1433	1431	0.47	C_15_H_24_
28	cis-Muurola-3,5-diene	1449	1448	0.42	C_15_H_24_
29	trans-Muurola-3,5-diene	1453	1451	0.56	C_15_H_24_
30	α-Humulene	1460	1452	0.66	C_15_H_24_
31	cis-Cadina-1(6),4-diene	1466	1461	0.31	C_15_H_24_
32	trans-Cadina-1(6),4-diene	1475	1475	0.83	C_15_H_24_
33	Germacrene D	1479	1480	1.64	C_15_H_24_
34	γ-Muurolene	1486	1478	10.63	C_15_H_24_
35	β-Selinene	1495	1489	1.19	C_15_H_24_
36	trans-Muurola-4(14),5-diene	1497	1493	0.88	C_15_H_24_
37	Bicyclogermacrene	1501	1500	2.50	C_15_H_24_
38	trans-β-Guaiene	1503	1502	0.29	C_15_H_24_
39	δ-Amorphene	1507	1511	0.001	C_15_H_24_
40	Germacrene A	1513	1508	0.03	C_15_H_24_
41	γ-Cadinene	1519	1513	0.68	C_15_H_24_
42	δ-Cadinene	1523	1522	4.22	C_15_H_24_
43	cis-Calamenene	1528	1528	1.16	C_15_H_22_
44	Myristicin	1537	1517	1.08	C_11_H_12_O_3_
45	α-Cadinene	1542	1537	0.33	C_15_H_24_
46	α-Calacorene	1550	1544	0.19	C_15_H_20_
47	ND	-	-	0.25	-
48	(*E*)-Nerolidol	1566	1561	0.76	C_15_H_26_O
49	Dodecanoic acid	1578	1565	0.87	C_12_H_24_O_2_
50	ND	-	-	0.33	-
51	ND	-	-	-	-
52	ND	-	-	-	-
53	Spathulenol	1590	1577	10.13	C_15_H_26_O
54	ND	-	-	1.17	-
55	Globulol	1597	1590	1.06	C_15_H_24_O
56	Salvial-4(14)-en-1-one	1605	1594	1.33	C_15_H_24_O
57	ND	-	-	0.74	-
58	10-epi-γ-Eudesmol	1619	1622	0.95	C_15_H_26_O
59	1,10-di-epi-Cubenol	1625	1618	0.69	C_15_H_26_O
60	Allo-aromadendrene epoxide	1629	1639	1.31	C_15_H_24_O
61	ND	-	-	0.28	-
62	1-epi-Cubenol	1639	1627	1.51	C_15_H_26_O
63	Cedr-8(15)-en-9-α-ol	1646	1650	0.88	C_15_H_24_O
64	epi-α-Cadinol	1654	1638	1.88	C_15_H_26_O
65	epi-α-Muurolol	1657	1640	1.25	C_15_H_26_O
66	Cubenol	1660	1645	1.51	C_15_H_26_O
67	ND	-	-	0.25	-
68	α-Cadinol	1669	1652	3.58	C_15_H_26_O
69	Germacra-4(15),5,10(14)-trien-1α-ol	1674	1680	1.12	C_15_H_24_O
70	ND	-	-	0.53	-
71	ND	-	-	0.26	-
72	5-Cyclodecen-1-ol	1692	1694	0.49	C_15_H_24_O
73	ND	-	-	0.40	-
74	Eudesma-4(15),7-dien-1β-ol (impure)	1706	1687	2.21	C_15_H_24_O
75	ND	-	-	0.28	-
76	14-hydroxy-α-Muurolene	1771	1779	0.57	C_15_H_24_O
77	Drimenol	1788	1766	1.92	C_15_H_26_O
		MH		17.199	
		MO		4.18	
		SH		32.27	
		SO		33.17	
		Others		1.95	
		Non identified		4.501	
		Total identified		93.27	

Note. CRI: calculated retention indexes; LRI: literature retention indexes [14]; MF: molecular formula; MH: hydrocarbon monoterpenes; MO: oxygenated monoterpenes; SH: hydrocarbonated sesquiterpenes; SO: oxygenated sesquiterpenes; NI: not identified.

**Table 2 plants-13-01806-t002:** Enantioselective analysis of D. Granadensis EO using a MEGA-DEX DET-Beta column.

Retention Time	Compound	CRI	LRI	Enantiomeric Distribution	Enantiomeric Excess e.e. (%)
3.912	(1S,5R)-(+)-α-pinene	944	935	16.49	77.02
3.949	(1S,5S)-(–)-α-pinene	946	943	93.51	
4.347	(1R,4S)-(–)-camphene	962	960	100.00	100.00
5.194	(1S,5R)-(+)-β-pinene	998	996	0.26	99.48
5.242	(1S,5S)-(–)-β-pinene	1000	999	99.74	

Note. RT: retention time; CRI: calculated retention indexes; LRI: literature retention indexes [15].

**Table 3 plants-13-01806-t003:** Antibacterial and antifungal capacities of *D. grandensis EO* against reference common pathogenic strains, measured as the minimum inhibitory concentration (MIC) and expressed in µg/mL.

Microorganisms	*S. rubricaulis Essential oil*	Antimicrobial Agent ^†^ (Positive Control)
Cocci Bacteria		Ampicillin
*Enterococcus faecalis*	1000	0.78
*Enterococcus faecium*	1000	<0.39
*Staphylococcus aureus*	500	<0.39
Rod-shaped Bacteria		Ciprofloxacin
*Lysteria monocytogenes*	250	1.56
*Escherichia coli* (O157:H7)	Non active	1.56
*Pseudomonas aeruginosa*	Non active	<0.39
*Salmonella enterica serovar Thypimurium*	Non active	<0.39
Microaerophile Rod-shaped bacteria		Erythromycin
*Campylobacter jejuni*	2000	15.63
Yeasts and sporulated fungi		Amphotericin B
*Candida albicans*	Non active	<0.098
*Aspergillus niger*	2000	<0.098

Non-active at the maximum dose tested of 2000 µg/mL. ^†^ MIC values were compared with quality controls (QC) reported in supplement M100 from CLSI.

**Table 4 plants-13-01806-t004:** *D. granadensis* essential oil antioxidant activity.

Sample	DPPH	ABTS	TEAC
	SC_50_ (µg/mL—µM *) ± SD		
*D. granadensis*	4181.74 ± 1.47	210.48 ± 1.03	325.54 ± 0.02
Trolox *	35.54 ± 1.04	29.09 ± 1.05	

* SC_50_: Half-scavenging capacity is expressed as µM for Trolox. TEAC: Trolox equivalent antioxidant capacity is expressed in µM per gram of EO.

## Data Availability

Data are contained within the article.

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
