# Peer review of "Chemical Profiling of Drimys granadensis (Winteraceae) Essential Oil, and Their Antimicrobial, Antioxidant, and Anticholinesterase Properties"

_plants, 2024, doi:10.3390/plants13131806_

Round 1

Reviewer 1 Report

Comments and Suggestions for Authors

The manuscript ID plants-3048907 is dedicated to determining and quantifying the chemical compounds in the essential oil of Drimys granadensis using a GC-MS method and some in vitro activities. However, from my perspective, the topic could be interesting for the scientific community, given that there are three publications in the literature. However, authors must mature the idea and results, reorient and thoroughly revise what they have written. As it stands, I reject the publication of Plants magazine.

Some comments and suggestions:

1. The summary does not invite you to continue reading. It is suggested to rewrite

2. Improve the introduction: there are only 6 references in the entire section and many claims lack scientific support.

3. The “Results and Analysis” and “Discussion” sections have the number “2”

4. The results of in vitro trials could be more encouraging. It is suggested to focus and deepen the enzymatic inhibition.

5. Update references (<40% in the last five years)

Comments on the Quality of English Language

English is correct

Author Response

Dear editor

A general document with all the suggestions made bye reviewers has been adressed and uploaded

Regards

Reviewer 2 Report

Comments and Suggestions for Authors

Plants-3048907: Chemical profiling of Drimys granadensis (Winteraceae) essential oil, and their antimicrobial, antioxidant, and anticholinesterase properties

The paper should be reviewed in some of its sections and improved in its presentation.

1-      Abstract

The MIC values ​​obtained are moderate or not very active, that should be changed in the abstract in the main text, and conclusions. This considering the MIC values ​​of the reference antibiotics.

The rules according to which the test is carried out should be mentioned here (CLSI or EUCASTAT or other)

2-      Abstract

Regarding phrase “Additionally, the oil exhibited strong anticholinesterase inhibitory potential, with an IC50 value of 63.88 ± 1.03 µg/mL.”

If compared to galanthamine which shows an IC50 1-2 µg/ml, the value reported here is moderate, this should be analyzed and modified.

Please see lines 149-152, page 7 section 2.5 Anticholinesterase potential

3-      Results

3.1- Table 3

Antimicrobial agent (Positive control)†

Please include only the MIC values ​​in µg/ml, the value of 1mg/ml is not understood? included for each antibiotic

4-      2.5 Anticholinesterase potential

Figure 3 is not necessary; it should be removed from the main text.

5-      Please see the following references on suggested MIC ranges to consider an extract or compounds with antimicrobial potential,  1-          and then review the paragraph between lines 218-230, page 9.

a-           For compounds, this stringent endpoints criteria were: significant (MIC<10μg/ml), moderate (10<MIC≤100 μg/ml) and low or negligible (MIC > 100 μg/ml)  Kuete V, Efferth T (2010) Cameroonian medicinal plants: pharmacology and derived natural products. Front Pharmacol 1:123”

b-           “Regarding edible plant extracts or their parts, it is estimated that they are very active if they show MIC values < 100 μg/mL, significantly active if 100 ≤ MIC ≤ 512 μg/mL, moderately active if 512 < MIC ≤ 2048 μg/mL and not very active if MIC > 2048 μg/mL (Tamokou et al., 2017).Medicinal Spices and Vegetables from Africa Therapeutic Potential Against Metabolic, Inflammatory, Infectious and Systemic Diseases2017, Pages 207-237 Medicinal Spices and Vegetables from AfricaChapter 8 - Antimicrobial Activities of African Medicinal Spices and Vegetables J.D.D.Tamokou, A.T.Mbaveng, V.Kuete,Tamokou, J.D.D., Mbaveng, A.T., Kuete

c-            Please see Rios J-L, Recio MC. Medicinal plants and antimicrobial activity. J Ethnopharmacol. 2005;100(1–2):80–4.

d- Liang, M., Ge, X., Xua, H., Ma, K., Zhang, W., Zan, Y., ... & Hua, X. (2022). Phytochemicals with activity against methicillin-resistant Staphylococcus aureus. Phytomedicine, 100, 154073.

6-      Regarding paragraph lines 240-243, page 9:

“The antifungal activity of drimenol against B. cinerea was evaluated and

recorded an EC50 between 40 to 80 ppm [17], unfortunately, this capacity could not be detected in our antifungal assay, probably to the low content of drimenol in our EO (1.91%)”

what does it mean? the phrase EC50 between 40 to 80 ppm

7-      4.3. Essential Oil Extraction

If possible include performance and some basic properties such as density

8-      4.5 Antimicrobial activity

Include the original document CLSI or EUCAST, and reference with which the test was developed.

9-      5. Conclusions

After the suggested changes, the conclusions must be reviewed and rewritten.

After the suggested changes, the manuscript could be considered for acceptance.

Comments on the Quality of English Language

Moderate editing of English language required

Author Response

(The authors gave the same response as above.)

Reviewer 3 Report

Comments and Suggestions for Authors

This study analyzed chemical compositions of essential oil extracted via steam distillation from dried Drimys granadensis leaves. Sixty-four chemical compounds, constituting 93.27% of the total composition, were identified, with major components including γ-Muurolene (10.63%), Spathulenol (10.13%), Sabinene (5.52%), and δ-Cadinene (4.22%). And their antimicrobial, antioxidant, and anticholinesterase properties were also accessed. In general, the experiments were well-performed and the manuscript was well-written. It may be published pending some minor revisions.

1) Not all abbreviations have been introduced with full names, for example, MIC, DPPH, ABTS, TEAC and IC50 in Abstract. And the full name should follow the first appearance of the abbreviation immediately, e.g. "GC/MS" in Line 20.

2) The Latin names, which appear for the first time, should be in full names, such as L. monocytogenes and S. aureus in Lines 29-30.

3) The Table 3 is hard to be understood. "1000" of what? Why to list the positive controls? "Ampicillin (1 mg/mL)" means its IC50 for Enterococcus faecalis was 1 mg/mL?

4) The unit for Figure 3 is missing. What does the dotted line mean?

Author Response

(The authors gave the same response as above.)

Round 2

Reviewer 2 Report

Comments and Suggestions for Authors

The authors have considered the suggestions made. The manuscript should now be accepted in its current state.

Comments on the Quality of English Language

Minor editing of English language required

Author Response

Comment 1: Minor editing of English language required

The text looks good! Here's a slightly improved version:

Dear Editor,

Thank you very much for your response. We have corrected the minor errors you suggested, and we have also extensively reviewed the main text to eliminate any language misspellings, both scientific and non-scientific.

Regards,